# Sol-Gel-Derived Fibers Based on Amorphous *α*-Hydroxy-Carboxylate-Modified Titanium(IV) Oxide as a 3-Dimensional Scaffold

**DOI:** 10.3390/ma15082752

**Published:** 2022-04-08

**Authors:** Bastian Christ, Walther Glaubitt, Katrin Berberich, Tobias Weigel, Jörn Probst, Gerhard Sextl, Sofia Dembski

**Affiliations:** 1Fraunhofer Institute for Silicate Research ISC, Translational Center Regenerative Therapies TLC-RT, Neunerplatz 2, 97082 Würzburg, Germany; walther.glaubitt@isc.fraunhofer.de (W.G.); katrinberberich79@gmail.com (K.B.); tobias.weigel@isc.fraunhofer.de (T.W.); joern.probst@isc.fraunhofer.de (J.P.); gerhard.sextl@isc.fraunhofer.de (G.S.); sofia.dembski@isc.fraunhofer.de (S.D.); 2Department Chemical Technology of Material Synthesis, University Würzburg, Röntgenring 11, 97070 Würzburg, Germany; 3Department Tissue Engineering and Regenerative Medicine, University Hospital Würzburg, Röntgenring 11, 97070 Würzburg, Germany

**Keywords:** sol-gel chemistry, scaffold, dry spinning

## Abstract

The development of novel fibrous biomaterials and further processing of medical devices is still challenging. For instance, titanium(IV) oxide is a well-established biocompatible material, and the synthesis of TiO_x_ particles and coatings via the sol-gel process has frequently been published. However, synthesis protocols of sol-gel-derived TiO_x_ fibers are hardly known. In this publication, the authors present a synthesis and fabrication of purely sol-gel-derived TiO_x_ fiber fleeces starting from the liquid sol-gel precursor titanium ethylate (TEOT). Here, the *α*-hydroxy-carboxylic acid lactic acid (LA) was used as a chelating ligand to reduce the reactivity towards hydrolysis of TEOT enabling a spinnable sol. The resulting fibers were processed into a non-woven fleece, characterized with FTIR, ^13^C-MAS-NMR, XRD, and screened with regard to their stability in physiological solution. They revealed an unexpected dependency between the LA content and the dissolution behavior. Finally, in vitro cell culture experiments proved their potential suitability as an open-mesh structured scaffold material, even for challenging applications such as therapeutic medicinal products (ATMPs).

## 1. Introduction

Titanium(IV) oxide has been an established implant material for decades, with high biocompatibility [1]. This biocompatibility of pure titanium or titanium alloys for hip or tooth implants is related to the formation of a thin, passivating non-stoichiometric Ti_x_O_y_ layer on the surface [2,3]. The layer is generated within seconds of contact with air or water [4]. Titanium(IV)-oxide-based materials are not only used in the replacement of hard tissue [5] but also in the regeneration of soft tissue, for instance, as a hernia mesh [6,7].

Metal alkoxides can be processed via sol-gel technology into differently shaped metal-oxide-based materials such as particles, coatings, or fibers [8,9,10,11,12]. However, the sol-gel reactions of titanium alkoxides are limited due to the fact of their high hydrolysis rate caused by the low electronegativity of Ti and the natural tendency to coordinate in a coordination number of eight [13]. Accordingly, titanium alkoxides directly react to amorphous titanium(IV) oxide in the presence of water. Therefore, it is challenging to process a material with a defined shape, such as monodisperse particles, fibers, or homogenous coatings via the sol-gel-technique. The addition of chelating ligands results in a decrease in the hydrolysis rate of titanium alkoxides. Thus, sol-gel processes can be performed and steered in a controlled way [14]. Meanwhile, many chelating ligands are known, for instance, diketones and diamines but also carboxylic acids to synthesize monodispersed TiO_x_ particles and coatings [15,16,17]. Particularly, in the fabrication of sol-gel-based fibers, a controlled reaction to a defined sol viscosity is of great importance and at the same time poses a great challenge in its fabrication [18]. Before gel formation, the hydrolysis and condensation reactions must be stopped to guarantee a fluid with a defined viscosity. These solutions can be spun to filaments by dry spinning [19], centrifugal spinning [20], or electrospinning techniques [21,22], while the evaporation of solvents turns the resulting fibers into a solid material.

In contrast to the common approach to doping organic polymeric filaments with crystalline rutile or anatase nanoparticles to refine the fiber properties [23,24,25,26], the manufacturing of amorphous filaments consisting solely of titanium-oxo-carboxylates is a novel material approach for TiO_x_-based filaments and opens new perspectives in material science.

For example, in tissue engineering as well as in regenerative medicine, plenty of fibrous scaffold materials are known [27], e.g., nanofibers mimic the extracellular matrix and promote cell growth and proliferation [28]. Concerning the *µ*m dimension, *µ*–fibers serve—in the form of single fibers or non-woven fabrics—as a fibrous guiding structure for the (re)growth of nerves [29,30] or blood vessels [31]. Fabricated to fiber fleeces, they are applied as three-dimensional scaffold structures for the regeneration of tissues. Apart from organic degradable biopolymers, such as polylactide, polyglycolide, or polycaprolactone, sol-gel-derived biocompatible inorganic fibers are also established. For instance, silica gel fibers, produced by the acidic catalyzed hydrolysis and condensation of tetraethoxysilane [19], represent a well-tolerated wound matrix in the regeneration of chronic venous leg ulcers [32] and, additionally, show anti-inflammatory and anti-fibrotic action in the healing of chronic wounds [33]. Furthermore, they promote cellular migration on their surface structures [34]. While currently known inorganic sol-gel fiber systems are mainly based on silicon, titanium-based sol-gel-materials are a promising option to expand the current field of inorganic material for challenging (biomedical) applications ahead.

For the first time, the authors present titanium(IV) oxide-based fibers produced via the sol-gel route using the liquid precursor titanium ethylate (TEOT) without any additional polymeric organic compounds. The fibers were spun via a dry spinning technique, fabricated into a non-woven fabric and evaluated as a biocompatible scaffold structure in contact with human dermal fibroblasts (hdf).

## 2. Materials and Methods

### 2.1. Materials

For fiber synthesis, TEOT (98%) was purchased from Dorf Ketal (Mumbai, India) and absolute ethanol (EtOH) as well as *D*,*L*-lactic acid (LA, 89%) from Sigma–Aldrich (St. Louis, MO, USA).

### 2.2. Sol Synthesis and Fiber Spinning

In a 2 L flask, 1 mol TEOT was mixed under stirring with 5 mol ethanol, and 1 mol (**1**) or 0.25 mol (**2**) LA was poured into the mixture at room temperature (RT). After 4 h, 18 mol (**1**) or 0.10 mol (**2**) deionized water was added to the clear yellow solution. Table 1 summarizes the sol compositions for the fabrication of two fiber materials **1** and **2**. Eighteen hours after pouring LA into the flask, solvents were removed by rotary evaporation (40 °C water bath temperature, 140–80 mbar reduced pressure) to a theoretical TiO_2_ solid content (theoretical assumption: full hydrolysis and condensation of TEOT into TiO_2_) of 30%. The resulting viscous sol was filled into a pressure container sealed with a nozzle plate with 7 nozzles with a diameter of 150 µm and cooled down to 15 °C. With a pressure of 20 bar, the viscous sol was pressed through the nozzles by synthetic air. Due to the high surface enlargement during filament formation, the residual solvents evaporated and solid filaments were obtained and assembled into a non-woven fleece.

### 2.3. Degradation Studies

Fibers were added into phosphate-buffered saline (PBS) in a ratio of 60 mg fibers to 1 L PBS. The PBS was changed every 2–3 days. To obtain a degradation profile, the fibers were washed with deionized water, dried in a vacuum oven at 40 °C, and the residual fiber weight was determined (n = 3).

### 2.4. Analytical Methods

#### 2.4.1. Scanning Electron Microscopy (SEM) and Energy-Dispersive X-ray Spectroscopy (EDX)

Fiber samples were prepared on a conductive carbon pad. The fiber surface was sputtered with Pt using a MED01 device of the company Balzers Union (Balzers, Liechtenstein) with 30 mA and a distance of 9 cm. SEM images were detected in a Supra^®^ 25 of the company Zeiss (Jena, Germany) by an InLens detector. The acceleration voltage was set to 3 kV and the sample detector distance to 3 mm. The EDX spectra were recorded using an EDAX element instrument applying an accelerating voltage of 10 kV, a working distance of 8.5 cm, and a measuring time of 200 s. The peaks were analyzed by an APEX software from EDAX (Mahwah, NJ, USA).

#### 2.4.2. Fourier Transform Infrared Spectroscopy (FTIR)

Two milligrams of a sample were ground into 300 mg of water-free KBr and pressed into a transparent pellet (10 tons for 5 min). The FTIR spectra were recorded in a wavenumber area between 4000 and 400 cm^−1^ and a scanning rate of 16 scans by using a FTIR-4100 spectrometer from the JASCO company (Tokyo, Japan) in the transmission mode.

#### 2.4.3. X-ray Powder Diffraction (XRD)

X-ray diffraction patterns were recorded on a STOE Stadi P instrument (Darmstadt, Germany) with a Cu-Kα radiation source (λ = 1.541 Å) and a curved Ge (111) monochromator. The measurements were carried out in the range of 2–40° 2*θ* with a step size of 0.5° and a measurement duration of 30 s in transmission geometry. The sample was prepared between two sheets of polyacetate.

#### 2.4.4. ^13^C Magic Angle Spinning Nuclear Magnetic Resonance Spectroscopy (^13^C-MAS-NMR)

The ^1^H decoupled ^13^C-VACP/MAS-NMR spectra were recorded with a frequency of 100.6 MHz and a sample amount of 300 mg on a DSX-400 from Bruker (Billerica, MA, USA). SiMe_4_ (δ = 0 ppm) was used as an external standard.

#### 2.4.5. Thermogravimetric Analysis (TGA)

TGA was performed using a TG209 IRIS instrument from Netzsch (Selb, Germany). The sample was heated from 30 °C up to 900 °C with a constant heating rate of 10 K/min under synthetic air conditions.

#### 2.4.6. ^1^H- and ^13^C Nuclear Magnetic Resonance Spectroscopy (NMR)

One hundred milligrams of **2** were incubated in 6 mL of D_2_O for 6 h. The supernatant was filtered, and NMR spectra were recorded on a Bruker DRX 500 instrument (Billerica, MA, USA) at room temperature. The NMR tube was stored at room temperature for 7 d and the NMR measurement was repeated.

### 2.5. In Vitro Cell Culture Experiments

#### 2.5.1. WST-1-Assay

The mitochondrial activity of the cell line, L929 (ATCC, Manassas, VA, USA), in contact to degradation products of **2** was tested in triplicates in 12-well plates on subconfluent monolayers by WST-1 reagent. One hundred milligrams of **2** were incubated in 1 mL of cell culture medium (DMEM-Glutamax^®^, Gibco, life technologies^TM^, Carlsbad, CA, USA) for 24 h. Afterwards, the supernatant plus 10% FCS (Gibco, life technologies^TM^, Carlsbad, CA, USA) was incubated on L929 cells in an undiluted form as well as 50% and 25% dilutions with DMEM +10% FCS (n = 3). For positive and negative controls, 10% sodium dodecyl sulfate (SDS, Carl Roth GmbH, Karlsruhe, Germany) and fresh DMEM with 10% FCS were applied. All cells were cultivated for a further 24 h, until 1 mL of WST-1 reagent was added in a 1:10 dilution in PBS+ to each well. After 30 min incubation time, 200 µL of each well were transferred into a transparent 96-well plate and analyzed by an ELISA reader (Infinte^®^ M200, Tecan, Männedorf, Switzerland) at a wavelength of 450 nm in absorption mode.

#### 2.5.2. Immunofluorescence Stainings and Fluorescence Imaging

For direct cell contact, primary hdf were applied. Hdf isolation was performed as described previously [35]. For analyzing cells in direct contact to **2**, 100 µL of a hdf suspension in DMEM–Glutamax^®^ plus 10% FCS (50,000 cells/100 µL) were seeded on the non-woven fabric. After 1 h of incubation (37 °C, 5% CO_2_), a further 900 µL of DMEM/FCS-mixture was added to each well. The culture was carried out for 3 d. Next, cells were washed with PBS and fixed for 10 min with Roti^®^HistoFix (Carl Roth GmbH). In advance of anti-vimentin staining, the fixed cells were treated with 0.2% Triton-X100 (Sigma–Aldrich) in PBS for 5 min, washed with PBS + Tween-20 (0.5%, Sigma–Aldrich) for 5 min, and blocked with 5% donkey serum for 20 min. Subsequently, primary rabbit-anti-vimentin antibodies (1:1000, abcam, Cambridge, UK) were incubated over night at 4 °C. After washing, secondary donkey-anti-rabbit-488 antibodies (1:200, Invitrogen, Waltham, MA, USA) were incubated for 1 h. Finally, samples were washed again and mounted with Mowiol-DAPI (Southern Biotech, Birmingham, AL, USA). Imaging was performed either with a fluorescence microscope (BZ-9000, Keyence; vimentin: GFP-Filter cube: excitation 470 nm, emission 535 nm; DAPI-filter cube: excitation 360 nm, emission 460 nm) or a confocal microscope (LSM SP8, Leica; vimentin: laser excitation: 488 nm, emission filter: 474–487 nm; DAPI: laser excitation: 405 nm, emission filter: 417–471 nm).

## 3. Results and Discussion

In sol synthesis, ethanol was used as a solvent to prevent transesterifications of TEOT [14], and a metabolizable acid—lactic acid (LA) as an α-hydroxy carboxylic acid—was used as a chelating agent to generate a non-cytotoxic material, after the addition of water. A sol was prepared (**1**: M(TEOT:LA:H_2_O:EtOH) = 1:1:18:5) in a flask, stirred for 18 h at RT, and evaporated off to a viscous sol with a solid content of 30%. The viscous sol was pressed by air pressure through nozzles (diameter: 150 *µ*m). The accompanied surface enlargement during filament formation resulted in the evaporation of residual solvents, and solid fibers were collected after a 3 m fall distance (Figure 1a). The obtained colorless and intrinsically stable endless fibers (**1**) were fabricated into a non-woven fiber fleece (Figure 1b).

Imaging of the fibers by SEM demonstrated a smooth surface and a fiber diameter of approximately 30 *µ*m (Figure 1c).

By analyzing the material structure, the XRD of mortared fibers showed an amorphous material, as no reflexes were obtained in a range of 10–40° 2*θ*. After heating the sample to 900 °C under an air atmosphere, XRD patterns showed (110), (101), and (111) of rutile, an crystalline form of titanium(IV) oxide (see Appendix A).

The coordination mode of LA to Ti in **1** was determined by FTIR. The difference *Δ* in the asymmetric *ν*_as_(COO^−^) and the symmetric *ν*_s_(COO^−^) stretching mode of the coordinated carboxylate group indicates a monodentate or bidentate coordination of the carboxylate to a metal atom [36]. A monodentate chelation (related to the carboxylate group) results in a greater *Δ* than in uncoordinated carboxylate [37]. In the case of the *α*-hydroxy carboxylic acids, the monodentate coordination mode formed a five-membered ring via the *α*-hydroxy-group. Bidentate chelation (*Δ* = 120–160 cm^−1^) is known for titanium–oxo–carboxylate complexes to be bridged over two titanium atoms forming a six-membered ring (Figure 2a) [38]. A FTIR-spectrum (Figure 2b) of uncoordinated LA (85%) shows, among others, a deformation vibration band of water *δ*(H_2_O) at 1638 cm^−1^ [39] and an asymmetric *ν*_as_(COO^−^) as well as a symmetric stretching mode *ν*_s_(COO^−^) at 1729 and 1405 cm^−1^ with a *Δ* of 324 cm^−1^. The spectrum of **1** showed two bands for *ν*_s_(COO^−^) as well as for *ν*_as_(COO^−^). The band positions of *ν*_s_(COO^−^) could be identified at 1310 and 1419 cm^−1^. One band for *ν*_as_(COO^−^) could be clearly located at 1568 cm^−1^. The second band of *ν*_as_(COO^−^) was overlapped by *δ*(H_2_O) from solvent residues in the fiber matrix, and it was positioned at approximately 1649 cm^−1^. No vibration bands of non-chelated LA were detected. The *Δ* of 149 and ~340 cm^−1^ were obtained and resembled a bidentate and monodentate coordination mode for the carboxylate group [39,40].

To further specify the coordination modes, the fibers were analyzed by solid-state ^13^C-MAS-NMR-spectroscopy (Figure 2c). Two peaks at 189.1 and 183.0 ppm can be assigned to C-atoms in a carboxylate group coordinated to Ti; the peaks at 84.2 and 69.1 ppm corresponded to a C-atom in the *α*-hydroxy position and the peak at 20.6 ppm to the methyl group [41]. The signal at 73.5 ppm together with the peak at 20.6 ppm indicates the presence of remaining ethoxy groups in the fiber matrix [42], which are not clearly identified by FTIR spectra.

Degradation studies of **1** in phosphate-buffered saline (PBS) show a full dissolution of fibers within seconds (see Appendix A). To the knowledge of the authors, formulations of water-soluble α-hydroxy-carboxylate-modified titanium complexes via hydrolysis and condensation of titanium alkoxides in the same molar ratio between Ti and the chelating acid have not yet been published. All water soluble titanium–carboxylate complexes synthesized via a titanium alkoxide route are described as being composed of a Ti-to-chelating-carboxylate ratio < 1 [43]. Next to commercially available titanium(IV)bis(ammonium-lactato)dihydroxide-complex (TiBALDH; Ti:lactate = 1:2) [44], other water soluble complexes with *α*-hydroxy carboxylate ligands are synthesized only via a peroxy route starting from titanium powder. Here, the use of chelating agents, such as lactic acid, malic acid, tartaric acid, or citric acid, results in water soluble complexes without alkoxo-ligands [45]. Via this synthesis route, water soluble peroxo-complexes with a ratio of 1:1 (Ti:carboxylate) are known, e.g., [Ti_4_(citrato)_4_(O_2_)_4_][NH_4_] [46].

To be suitable for a scaffold structure, the LA content had to be reduced in the sol composition in order to increase the condensation grade of the hybrid fibers which, in turn, reduced the fiber solubility in aqueous media. A screening of different LA and water ratios with unaltered content of TEOT and EtOH was performed to obtain spinnable sols. Along with the reduction in chelate ligands and the resulting greater sensitivity of the sol to hydrolysis reactions, the water contents of the sols had to be reduced. Spinnable sols remained clear after the addition of water; non-spinnable sols showed an amorphous TiO_x_ precipitation (Figure 3a). For further examination, the sol with the smallest LA and water content was chosen for spinning (sol composition M(TEOT:LA:H_2_O:EtOH) = 1:0.25:0.1:5).

In SEM images, the resulting fibers (**2**) showed a smooth surface area and a diameter of 49.2 ± 2.3 µm measured on 10 single fibers by ImageJ. EDX measurements proved the presence of Ti within the as-spun filaments. Next to peaks of the elements with a low atomic number for C and O, clear K_α_ and K_β_ peaks at 4.511 and 4.931 keV were identified and can be assigned to the presence of Ti within the fiber matrix [47]. A TGA measurement of **2** under synthetic air conditions from RT up to 900 °C showed at 900 °C a value of 59.4 m%. Afterwards, the remaining, heat-treated sample was analyzed by XRD. The obtained reflexes can be assigned to the crystalline TiO_2_ modification rutile (110), (101), and (111) (see Appendix A) [48].

In the FTIR analysis, monodentate as well as bidentate coordination modes of LA were recorded. Clear bands of *δ*_as_(CH_3_) at 1448 cm^−1^ and *δ*_s_(CH_3_) at 1384 cm^−1^ of the remaining ethoxy groups were detected (see Appendix A) [49].

By a gravimetrical degradation study in PBS (Figure 3b), an increased stability in aqueous solutions was demonstrated, with a degradation of approximately 20 m% within 8 h. Afterwards, the fiber weight remained almost constant.

Solution ^1^H-NMR measurements of **2** that degraded in D_2_O for 6 h showed signals for lactic acid at 4.24 (quartet) and 1.31 ppm (doublet) [50]; ^13^C-NMR-spectra indicated peaks at 179.5, 66.9, and 19.4 ppm. By recording the NMR spectra of the same sample again after 7 days, new peaks appeared: a quartet at 4.97 ppm and a doublet at 1.26 ppm in the ^1^H-NMR spectrum as well as peaks in the ^13^C-NMR spectrum at 188.4, 82.2, and 19.9 ppm (Figure 4). These peaks can be assigned to the water-soluble and -stable mononuclear titanium complex coordinated by three LA ligands [41,44].

Thereby, an indirect proof for the release of titanium species out of the fibers is given. After all, TiO_2_ is known to release Ti(OH)_4_ or Ti(O)(OH)_2_ in aqueous solution until a saturation concentration of ~1 *µ*M is achieved [51]. This effect has been known in patients with titanium implants for many years. Here, titanium is bound to the Fe^III^-binding blood protein serum transferrin or forms water soluble complex Ti(citrate)_3_ in the presence of citrate in blood plasma [52]. Recently, an influence of titanium citrate species was identified in the differentiation process to mature adipocytes [53]. Due to the ubiquity of titanium in our environment, the uptake of titanium ions into the body is (even) unavoidable and Ti is present in every human [54]. After all, titanium is the fourth most common metal in the Earth’s crust [55], is omnipresent in seawater [56,57] and plants [58] and, in the form of synthetic crystalline particles, a commonly used white pigment [59]—even in cosmetics, tooth pastes, and sun protection [60].

To prove the applicability of titanium(IV)-oxide-based fleeces as novel scaffold structures in tissue engineering, the cytotoxicity of the degradable material was evaluated based on DIN ISO 10993-5. Here, 60 mg of **2** were incubated in 2 mL of cell culture media (DMEM) for 24 h. Afterwards, the DMEM, loaded with material extract, was incubated for further 24 h with cell line L929. Figure 5a shows the metabolic activity of the cells (WST-1-assay) against a non-diluted, 50% diluted, and 25% diluted material extract in DMEM. The results were normalized against a negative control (cells incubated with DMEM without material contact). All bars exceeded the critical level of 80%; therefore, the material can be classified as non-cytotoxic regarding the impact of the degradation products on cells’ metabolic activity.

To evaluate cell adhesion and proliferation of primary hdf in direct material contact, primary hdf were cultured for 3 days on **2**, which were fabricated into a three-dimensional non-woven scaffold. After fixation of the cells, cell nuclei (blue) and the type III intermediate filament vimentin of the cytoskeleton (green) were stained for immunofluorescence imaging (Figure 5b).

As a picture of at least five fiber layers recorded via confocal microscopy shows, all the fibers’ surface areas were covered by hdf, and the cells showed vital morphologies (Figure 5c). After cell seeding, hdf first attached to the lower fiber fleece regions. This region of the scaffold was densely populated with hdf. At the angles of the two crossed fibers, the hdf stretched themselves to the next less populated fiber and used the fibrous scaffold structure to migrate to the upper region. In this region, the fiber surface was less populated, and hdf show an elongated morphology, typical for fibroblasts.

In conclusion, sol-gel synthesis of novel titanium–oxo-based sols modified with LA ligands and its fabrication to a non-woven fabric are presented. The material itself showed an unexpected behavior regarding its solubility in aqueous media. A lactate degree stoichiometric to the content of Ti atoms (**1**) resulted in quick dissolution of the material. After reducing the lactate and water content in the sol, the obtained fibers (**2**) only slightly degraded in the physiologic solution. The open-mesh structure was proven to be non-cytotoxic, and exemplary proof was provided of its ability to work as a degradable scaffold material for the cultivation of hdf.

## Figures and Tables

**Figure 1 materials-15-02752-f001:**
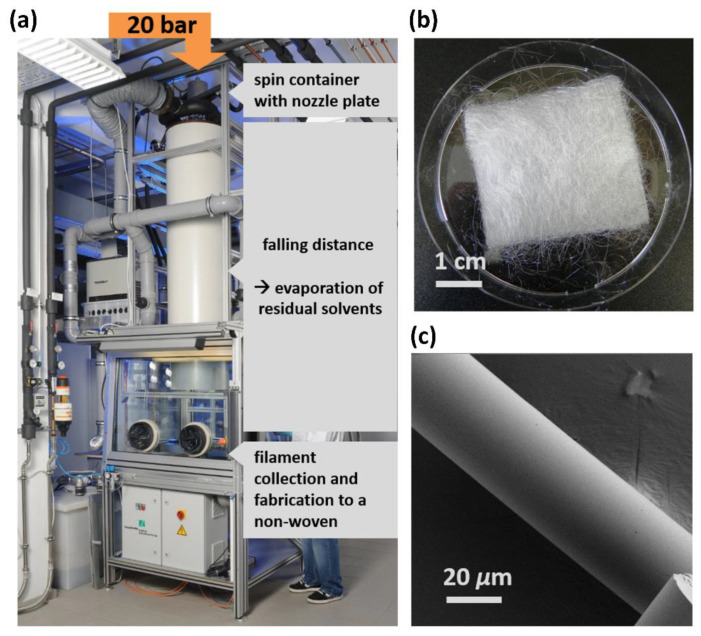
Picture of the spinning plant (**a**) and a non-woven fiber fleece **1** (**b**); SEM image of **1** with a fiber diameter of 30 µm and a smooth surface area (**c**).

**Figure 2 materials-15-02752-f002:**
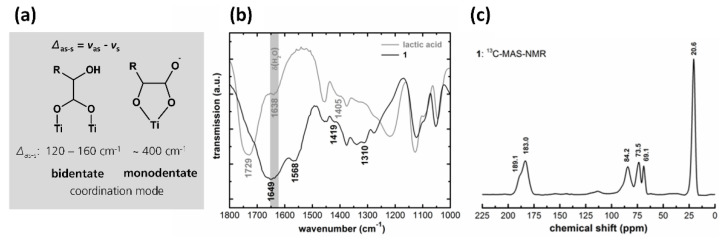
Differences (*Δ*_as-s_) in the asymmetric *ν*_as_(COO^−^) and the symmetric *ν*_s_(COO^−^) stretching mode of lactate chelated to Ti. Values of 120–160 cm^−1^ indicate a bidentate coordination of lactate; values >400 cm^−1^ indicate a monodentate coordination mode (**a**). The FTIR spectra of lactic acid (grey line) and **1** (black line). *ν*_as_(COO^−^) and *ν*_s_(COO^−^) are marked in the spectra indicating a monodentate as well as a bidentate coordination mode of lactate to Ti (**b**). The ^13^C-MAS-NMR spectrum of **1** indicating two different coordination modes of lactate to Ti (**c**).

**Figure 3 materials-15-02752-f003:**
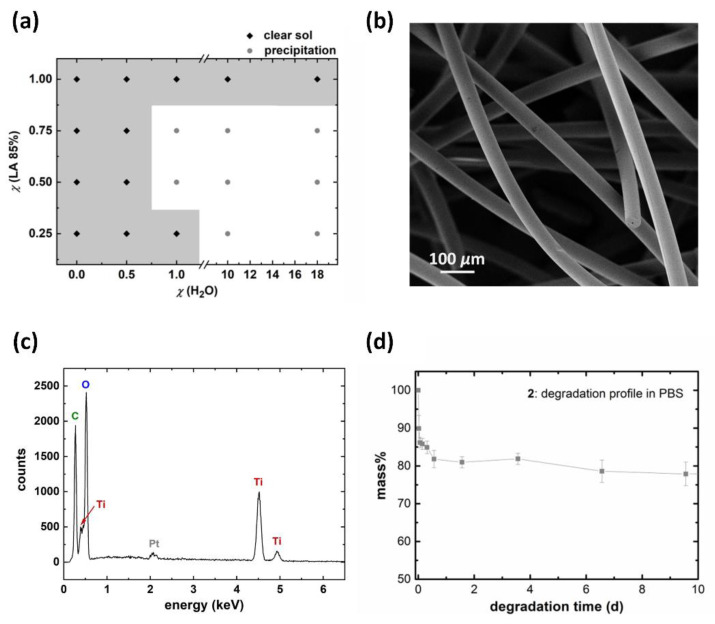
Variation of the LA and water ratio in sol resulted in clear spinnable sols (black rhombs) or colorless precipitates (grey circles) (**a**); SEM image of **2** assembled into a non-woven fleece (**b**); the EDX spectra of as-spun **2** (**c**) and a mass degradation profile of **2** in PBS over a period of 10 days. After an initial burst release the fibers hardly degrade (**d**).

**Figure 4 materials-15-02752-f004:**
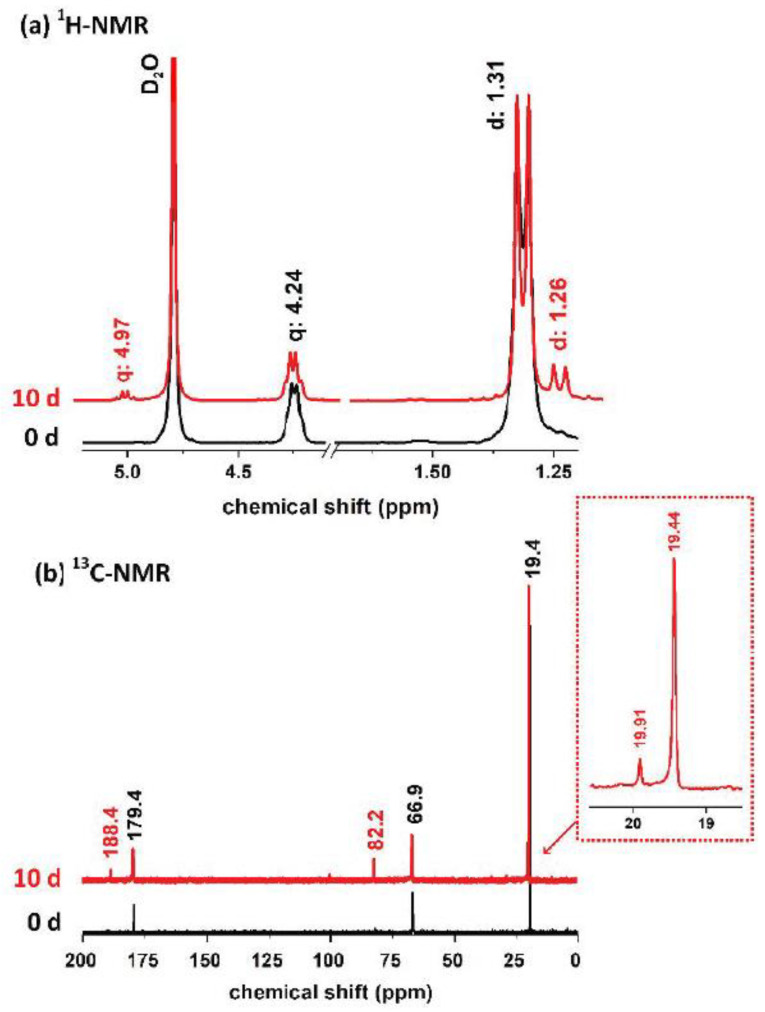
^1^H- (**a**) and ^13^C-NMR (**b**) spectra of dissolved degradation products of **2** after 6 h in D_2_O: (**black**) directly measured after dissolution experiment; (**red**) measured after 10 d stored in D_2_O containing an NMR tube. First, only signals for dissolved lactate were detected. After storage for 10 d, the signals for Ti–lactate species were additionally formed.

**Figure 5 materials-15-02752-f005:**
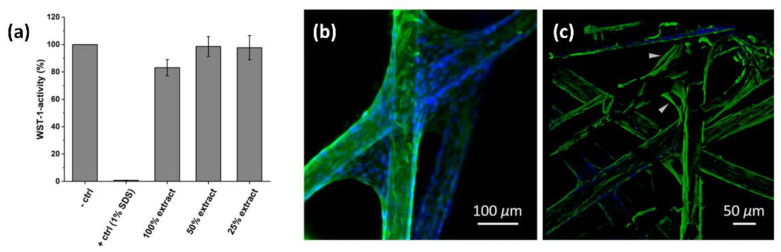
WST-1 activity of L929 cells after incubation with extracts of cell culture media in contact to **2** for 24 h (**a**); hdf seeded on **2** and stained for immunofluorescence imaging (blue: cell nuclei; green: type III intermediate filament vimentin of cytoskeleton, imaged by fluorescence microscopy) after a cultivation for 3 days (**b**). Thereby, hdf covered the complete fiber surface. To record a higher image volume, confocal microscopy was applied to reduce scattered light (**c**). At the angles of crossing fibers, hdf migrated from fiber to fiber (arrows) to grow all over the scaffold structure.

**Table 1 materials-15-02752-t001:** Overview of the sol compositions for the spinning of **1** and **2**.

	*n*(TEOT) (mol)	*n*(LA) (mol)	*n*(H_2_O) (mol)	*n*(EtOH) (mol)
**1**	1.00	1.00	18.00	5.00
**2**	1.00	0.25	0.10	5.00

## Data Availability

Data sharing not applicable.

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
