# Peer review of "Sol-Gel-Derived Fibers Based on Amorphous α-Hydroxy-Carboxylate-Modified Titanium(IV) Oxide as a 3-Dimensional Scaffold"

_materials, 2022, doi:10.3390/ma15082752_

Round 1
Reviewer 1 Report
The experimental design, characteristics data and the results are not well supported by the findings.
So, I cannot recommend it for publication.
The paper, titled "Sol-gel-derived fibers based on amorphous-hydroxy-carboxylate modified titanium(IV) oxide as a 3-dimensional scaffold," has been submitted for publication in the journal Materials. This research work is interesting. However, there are significant concerns regarding the novelty and the findings are not clear. I recommend the authors re-write the manuscript.
Specific comments
The authors claim that "the modified titanium (IV) oxide" is present as a 3-dimensional scaffold in the synthesized sample. However, there is no evidence in XRD and FTIR analysis for the presence of titanium (IV) oxide in the synthesized sample. The authors are advised to explain the SEM-EDS or XPS analysis for the evidence of the presence of titanium (IV) oxide.
Author Response
Comment 1: The experimental design, characteristics data and the results are not well supported by the findings.
So, I cannot recommend it for publication.
Response to comment 1: We thank the reviewer for his/her time and effort. We hope, that our following improvements in the manuscript can address the reviewers concerns and change his view on our manuscript.
Comment 2: The paper, titled "Sol-gel-derived fibers based on amorphous-hydroxy-carboxylate modified titanium(IV) oxide as a 3-dimensional scaffold," has been submitted for publication in the journal Materials. This research work is interesting. However, there are significant concerns regarding the novelty and the findings are not clear. I recommend the authors re-write the manuscript.
Response to comment 2: We thank the reviewer for this valuable comment. To improve the clarity as well as the scope of our manuscript, we extensively adapted the text in all parts of the manuscript.
Comment 3: The authors claim that "the modified titanium (IV) oxide" is present as a 3-dimensional scaffold in the synthesized sample. However, there is no evidence in XRD and FTIR analysis for the presence of titanium (IV) oxide in the synthesized sample. The authors are advised to explain the SEM-EDS or XPS analysis for the evidence of the presence of titanium (IV) oxide.
Response to comment 3: Thank you for this legitimate notice. Regarding scaffold 2, we added EDX spectra to qualitatively prove the titanium content of the sample. Additionally, we are presenting a thermogravimetric analysis (TGA) measurement, which, after heating to 900 °C under synthetic air atmosphere, has a residual mass of 59%. The heat-treated residue after TGA was examined by XRD. Here, reflexes of rutile were obtained. Thus, an evidence of the presence of titanium (IV) oxide based material is now shown.
Next to the XRD of as-spun 1, representing an amorphous material without reflexes in diffractogram, we also added an XRD of a heat-treated 1 (900 °C, air atmosphere). Here, (110), (101) and (111) reflexes of rutile were obtained that evidences the presence of TiOx species also in 1.
The methods TGA and EDX were added in the methods section.
Reviewer 2 Report
The manuscript “Sol-gel-derived fibers based on amorphous α-hydroxy-carboxylate modified titanium(IV) oxide as a 3-dimensional scaffold” describes an interesting topic but contains some incongruities. Therefore, it may be recommended for publication after minor revision:
1) Authors should complement the abstract and describe the purpose of the scientific work
2) Authors should describe in detail all results, in particular: insert statistical analysis
3) The following publication is recommended in introduction section:
- CATAURO, Michelina, et al. Antibacterial properties of sol–gel biomaterials with different percentages of PEG or PCL. In: Macromolecular Symposia. 2020. p. 1900056.
Author Response
The manuscript “Sol-gel-derived fibers based on amorphous α-hydroxy-carboxylate modified titanium(IV) oxide as a 3-dimensional scaffold” describes an interesting topic but contains some incongruities. Therefore, it may be recommended for publication after minor revision:
We are grateful for the reviewer’s positive feedback and the effort to improve our manuscript. We hope we could address the reviewer’s valuable comments sufficiently.
Comment 1: Authors should complement the abstract and describe the purpose of the scientific work
Response to comment 1: Thank you for this advice. We revised the abstract concerning the reviewers concerns.
Comment 2: Authors should describe in detail all results, in particular: insert statistical analysis
Response to comment 2: We thank the reviewer for this comment. After revision of this manuscript we added new analytic measurements, especially for fiber system 2 used in in-vitro experiments. Here, we also added a SEM image of 2 showing not only the smooth fiber surface and its diameter, but also its open-meshed scaffold structure to emphasize an applicability as a 3-dimensional scaffold. The fiber diameter was statistically analyzed by ImageJ.
To evidence the presence of Ti in 2, we enclosed EDX spectra to qualitatively prove the Ti content of the sample. Additionally, we present a thermogravimetric analysis (TGA) measurement of 2. After heating to 900 °C under synthetic air atmosphere, a residual mass of 59% remains. This heat-treated residue after TGA was examined by XRD. Here, reflexes of rutile were obtained. Thus, an evidence of a titanium (IV) oxide based material is now shown.
The methods FTIR, 13C-MAS-NMR, 1H- and 13C- liquid NMR, fluorescence microscopy, EDX are qualitative methods and do not allow quantification. As a result, these results are not statistically analyzed.
Experiments for the degradation profile and WST-1-assay were performed as triplicates. The standard deviation is shown in each degradation time or cell culture condition. We added this information to the related methods.
Comment 3: The following publication is recommended in introduction section:
- CATAURO, Michelina, et al. Antibacterial properties of sol–gel biomaterials with different percentages of PEG or PCL. In: Macromolecular Symposia. 2020. p. 1900056.
Response to comment 3: Thank you for this hint. We added this study as reference 4 in the introduction section.
Reviewer 3 Report
Line 165. How the authors measured 30% of solid contend? Or is the theoretical assumption?
Why do authors not present result of mix 2 in figure 1?
Author Response
We thank the reviewer for the feedback and the helpful comments to improve our manuscript, to which we would like to reply point by point below.
Comment 1: Line 165. How the authors measured 30% of solid contend? Or is the theoretical assumption?
Response to comment 1: We thank the reviewer for this important question. As mentioned in the methods section, the solid content of the sol is a theoretical assumption. “…to a theoretical TiO2 solid content (theoretical assumption: full hydrolysis and condensation of TEOT into TiO2) of 30 %.”
In detail, it is the quotient of the weight of the spinnable sol and the theoretical weight of a 100% hydrolyzed TEOT to a fully condensed to TiO2. Regarding the presence of TiOx within the sample we also enclosed a TGA measurement of 2 that evidences, after heating to 900 °C under synthetic air atmosphere, has a residual mass of 59%. The heat treated residue after TGA was examined by XRD. Here, reflexes of rutile were obtained. Thus, an evidence of the presence of titanium (IV) oxide based material is now also shown. A qualitative evidence of Ti of as-spun 2 is now also presented in an EDX spectra.
Comment 2: Why do authors not present result of mix 2 in figure 1?
Response to comment 2: Thank you for this legitimate comment. We decided to separate the different fibers, as only 2 is applicable for in-vitro studies. In respect, to the focus on cell culture, we also added a SEM image of 2 showing not only the smooth fiber surface and its diameter, but also its open-meshed scaffold structure to emphasize an applicability as a 3-dimensional scaffold. Next, we enclosed a TGA measurement of 2 that evidences, after heating to 900 °C under synthetic air atmosphere, has a residual mass of 59%. The heat treated residue after TGA was examined by XRD. Here, reflexes of rutile were obtained. Thus, an evidence of the presence of titanium (IV) oxide based material is now also shown.
Reviewer 4 Report
In the manuscript entitled “Sol-gel-derived fibers based on amorphous α-hydroxy-carboxylate modified titanium(IV) oxide as a 3-dimensional scaffold” the authors are reporting a new protocol used to prepare TiOx fibers used to obtain 3D biocompatible scaffolds.
The study presents in a concise manner the preparation and characterization of the fibers and finally in vitro studies were performed to show their biocompatibility.
Although the data are of interest, the manuscript needs improvement.
Through all the manuscript there are some phrases that need to be reformulated as the idea that authors want to present is not so clear.
In the material and methods, at 2.5 paragraph, the authors need to give a short title to the 2 paragraphs which are essentially describing 2 different methods. The same was done before, at 2.4.
Also, beside the name of the microscopes used, the authors should say the objective used to record the images, as well as the filter cubes.
In the results and discussion section the authors need to separate each results based on the methods used. It will make the manuscript easier to read.
For in vitro studies the authors need to say which of the two microscopes were used for the images presented.
Finally, the manuscript does not have a Conclusion section. The authors need to add it.
Author Response
In the manuscript entitled “Sol-gel-derived fibers based on amorphous α-hydroxy-carboxylate modified titanium(IV) oxide as a 3-dimensional scaffold” the authors are reporting a new protocol used to prepare TiOx fibers used to obtain 3D biocompatible scaffolds.
The study presents in a concise manner the preparation and characterization of the fibers and finally in vitro studies were performed to show their biocompatibility.
Although the data are of interest, the manuscript needs improvement.
We thank the reviewer for the detailed analysis of our manuscript and hope that we could sufficiently address the reviewer’s comments.
Comment 1: Through all the manuscript there are some phrases that need to be reformulated as the idea that authors want to present is not so clear.
Response to comment 1: Thank you for this advice. We extensively revised the language and formulations in the whole manuscript.
Comment 2: In the material and methods, at 2.5 paragraph, the authors need to give a short title to the 2 paragraphs which are essentially describing 2 different methods. The same was done before, at 2.4.
Response to comment 2: Thank you for this legitimate notice. We added the subheadlines “WST-1-assay” as well as “Immunofluorescence stainings and fluorescence imaging” in paragraph 2.5.
Comment 3: Also, beside the name of the microscopes used, the authors should say the objective used to record the images, as well as the filter cubes.
Response to comment 3: We thank the reviewer for this comment. We added the information concerning the applied filter cubes for the fluorescent microscope as well as the laser excitation and emission filter range of the confocal microscope in the methods part:
Imaging was performed either with a fluorescence microscope (BZ-9000, Keyence; vimentin: GFP-Filter cube: excitation 470 nm, emission 535 nm; DAPI-filter cube: excitation 360 nm, emission 460 nm) or a confocal microscope (LSM SP8, Leica; vimentin: laser excitation: 488 nm, emission filter: 474-487 nm; DAPI: laser excitation: 405 nm, emission filter: 417-471 nm).
Concerning the microscope objectives, the provision of objective magnifications is in most cases not representative to the displayed magnification. As the total magnification depends additionally on the second objective in the microscope, the size of the image varies on every computer display. Furthermore, the images are cropped to fit into the figure. In contrast, scale bars in the images are always representative to magnification. Therefore, we think the provision of objective magnifications would not improve microscopic image information.
Comment 4: In the results and discussion section the authors need to separate each results based on the methods used. It will make the manuscript easier to read.
Response to comment 4: We thank the reviewer for this comment. To improve the understanding and flow of understanding, we complied the reviewers suggestion and separated each results based on the methods used within the text.
Comment 5: For in vitro studies the authors need to say which of the two microscopes were used for the images presented.
Response to comment 5: We thank the reviewer for this comment. We added the requested microscope information into the related figure caption:
“hdf seeded on 2 and stained for immunofluorescence imaging (blue: cell nuclei; green: type III intermediate filament vimentin of cytoskeleton, imaged by fluorescence microscopy) after a cultivation for 3 days (b). Thereby hdf covered the complete fiber surface ; to record a higher image volume, confocal microscopy was applied to reduce scattered light (c). In the angles of crossing fibers, hdf migrated from fiber to fiber (arrows) to grow all over the scaffold structure. “
Comment 6: Finally, the manuscript does not have a Conclusion section. The authors need to add it.
Response to comment 6: Thank you for this legitimate notice. We added a conclusion section:
“In conclusion, sol-gel synthesis of novel titanium-oxo-based sols modified with LA ligands and its fabrication to a non-woven fabric are presented. The material itself showed an unexpected behavior regarding its solubility in aqueous media. A lactate degree stoichiometric to the content of Ti atoms (1), resulted into a quick material dis-solution. After reducing the lactate content in the sol, obtained fibers (2) are only slightly degrading in physiologic solution. The open-mesh structure was proven being non cytotoxic and exemplary proofed to work as a degradable scaffold material for the cultivation of hdf.”
Round 2
Reviewer 1 Report
Can be Accepted in present form.
Reviewer 4 Report
The authors have made the changes asked and the quality of the manuscript improved. The paper can be accepted in the present format.